# Rapid and biased evolution of canalization during adaptive divergence revealed by dominance in gene expression variability during Arctic charr early development

Quentin Jean-Baptiste Horta-Lacueva [1,2✉], Zophonías Oddur Jónsson [1], Dagny A. V. Thorholludottir[1,3], Benedikt Hallgrímsson [4] & Kalina Hristova Kapralova [1,5✉]

Adaptive evolution may be influenced by canalization, the buffering of developmental processes from environmental and genetic perturbations, but how this occurs is poorly understood. Here, we explore how gene expression variability evolves in diverging and hybridizing populations, by focusing on the Arctic charr (*Salvelinus alpinus*) of Thingvallavatn, a classic case of divergence between feeding habitats. We report distinct profiles of gene expression variance for both coding RNAs and microRNAs between the offspring of two contrasting morphs (benthic/limnetic) and their hybrids reared in common conditions and sampled at two key points of cranial development. Gene expression variance in the hybrids is substantially affected by maternal effects, and many genes show biased expression variance toward the limnetic morph. This suggests that canalization, as inferred by gene expression variance, can rapidly diverge in sympatry through multiple gene pathways, which are associated with dominance patterns possibly biasing evolutionary trajectories and mitigating the effects of hybridization on adaptive evolution.

[1] Institute of Life and Environmental Sciences, University of Iceland, Reykjavík, Iceland. [2] Department of Biology, Lund University, Lund, Sweden. [3] University of Veterinary Medicine Vienna, Institute of Population Genetics, Vienna, Austria. [4] Department of Cell Biology and Anatomy, Alberta Children's Hospital Research Institute, University of Calgary, Calgary, Alberta, Canada. [5] The Institute for Experimental Pathology at Keldur, University of Iceland, Reykjavík, Iceland. ✉email: quentin.horta-lacueva@biol.lu.se; kalina@hi.is

Extensive research efforts have been dedicated over the past few years to understand the evolution of ontogenetic differences between populations, mainly with the aim of unraveling the processes of adaptive divergence and speciation[1–11]. However, significant knowledge gaps remain. One of these is the potential role of canalization during speciation events[12–14]. Canalization refers to the relative reduction of genetic or environmental effects on phenotypic variation[15]. The genetic and evolutionary-developmental bases for canalization remain poorly understood[12,14,15], but there are strong reasons to expect the microevolutionary processes involved in speciation to modulate canalization. Existing work on this question has tended to focus on phenotypic variation[16–18]. Here, we investigate the impact of rapid phenotypic diversification and hybridization on gene expression variance using Arctic charr morphs (*Salvelinus alpinus*) from Thingvallavatn, Iceland.

In what way would strong directional selection and rapid morphological evolution impact canalization? Directional selection can increase homozygosity[19], thereby reducing genetic variance, but this may not result in a proportionate decrease in phenotypic variance. Instead, selection may increase phenotypic variance or even decrease developmental stability[20,21]. Why this occurs is not known, yet the effects of increased homozygosity on canalization or developmental stability have been associated with inbreeding depression due to the expression of recessive deleterious effects[22,23]. This is broadly consistent with Lerner's hypothesis of genetic homeostasis[24]. Alternatively, directional selection may also disrupt developmental configurations that have been shaped by stabilizing selection. This is consistent with the selection, pleiotropy and compensation model proposed by Pavlicev and Wagner[25] in which directional selection is proposed to result in deleterious pleiotropic effects that are subsequently ameliorated by compensatory selection—note, however, that empirical studies rather show that synergistic (i.e., positive) pleiotropy is important in adaptive evolution[26,27]. Rapid evolution, under this model, would be associated with increased variance due to deleterious pleiotropic effects. Finally, directional selection may increase phenotypic variance due to nonlinearities in genotype–phenotype maps for key trait-influencing genes. Such nonlinear maps can result in genetic variation for canalization (phenotypic variance within a genotype in this case)[28,29]. These nonlinearities are common, so directional selection might shift gene expression to steeper locations along gene expression to phenotype curves, such that the same amount of genetic variation would result in an increased amount of phenotypic variation.

Besides the effects of pleiotropy and nonlinear genotype–phenotype maps on phenotypic variation, canalization may evolve in unclear ways because populations have to undergo gene flow, at least during early divergence[24,30]. The effects of hybridization on phenotypic variance can go in either direction as revealed by studies of introgressing populations[17,31–33]. This is presumably because some hybridization events may result in the breakdown of genetic configurations that have undergone stabilizing selection, while others may enhance hybrid vigor through increased heterozygosity[34]. The importance of such mechanisms and how their effects on phenotypic variance condition to the levels of introgression and to the extent of genetic differentiation (i.e., genetic distances and nature of incompatibilities between populations) remain to be established.

Further complexity arises when considering trait dominance and parental effects—hereafter referred to as dominance in a broad sense[35–38]. For many traits for example, hybrids of recently diverged populations resemble one of the parents instead of being intermediate, potentially generating trait mismatches that complicate predictions about post-zygotic isolation[39]. Consequently, it becomes crucial to not only understand how dominance affects developmental processes which induce the divergence of average trait values, but also how dominance influences phenotypic robustness. Likewise, it is of special importance to consider the evolutionary dynamics of phenotypic robustness in the face of gene flow. This can be achieved by (1) investigating whether dominance affects phenotypic robustness in the same way as it affects average trait values, and (2) assessing its consequences for the maintenance of phenotypic variation between diverging populations.

Here, we studied two Arctic charr morphs that we know exhibit extensive morphological divergence as well as differences in phenotypic plasticity and phenotypic variance[40–42], and focused on the modulation of the abundance and the variance of gene expression between the two morphs as well as in their hybrids. Using gene expression as a molecular phenotype has the advantage of enabling analyses of a whole spectrum of traits (free from selection bias) instead of a focus on specific morphological characters[43–46]. Further, this level of analysis allows focusing on mechanistic questions related to genetic pathways and gene regulatory networks[47,48]. While our understanding of how variation in gene expression modulates development at the single-cell levels is growing[49,50], little has been achieved for whole organisms. Meanwhile, the recent advent of the -omics has enabled estimating transcriptome-wide differential expression in a wide range of organisms[51–53], but the variability in such expression remain virtually unexplored. We focused on the expression variability of both coding and noncoding RNAs. We specifically looked into microRNAs (miRNAs), which are major regulatory elements known for reducing the expression noise of target mRNAs[54–56]. In this paper, we refer to *variability* as the *tendency* of a system (e.g., an organism) to produce variation (e.g., in a phenotype), and to *variation* as the observable outcome of this phenomenon[15].

Sympatric Arctic charr morphs present classic and well-characterized cases of resource polymorphisms. In Thingvallavatn, two of the four described morphs constitute genetically differentiated populations despite wide overlaps in spawning time and location[57–61]: the planktivorous charr (PL), a limnetic morph, feeds on zooplankton and emerging chironomids, whereas the small benthic charr (SB) forages on benthic invertebrates within the lava matrix. The two morphs have contrasting body and head shapes, with clear morphological differences in the trophic apparatus (Fig. 1a, b), and occupy highly specialized niches[62]. However, the benthic morph is seemingly more morphologically derived than the limnetic morph when compared to the anadromous ancestor[40]. The two morphs are reproductively isolated[59], F₁ hybrids are rare in the wild[63], and recent studies on dominance in morphology and behavioral traits suggest that post-zygotic isolation might have evolved between the two morphs through trait mismatches[18,64,65]. These large-scale phenotypic changes have occurred rapidly (within 11,000 years), which strongly suggests rapid divergence under some combination of directional selection, drift, and founder effect[57]. These characteristics make this an ideal model to investigate the impact of microevolutionary processes on canalization at the level of gene expression variance.

The two charr morphs exhibit complex differences in morphological plasticity and in variance reduction of head and body shape over ontogeny[18,40], which may indicate intricate differences in canalization. Here, we estimated mRNA and miRNA expression variability in embryos of the benthic and the limnetic morphs and their reciprocal hybrids reared in common garden conditions. We focused on a timeframe when cartilage components of the feeding apparatus are developing[66] and shape differences between morphs start to appear[66]. First, if canalization rapidly evolves, we predicted that the two recently diverged morphs would differ in gene expression variability (measured as gene expression variance). Second, if hybridization relaxes

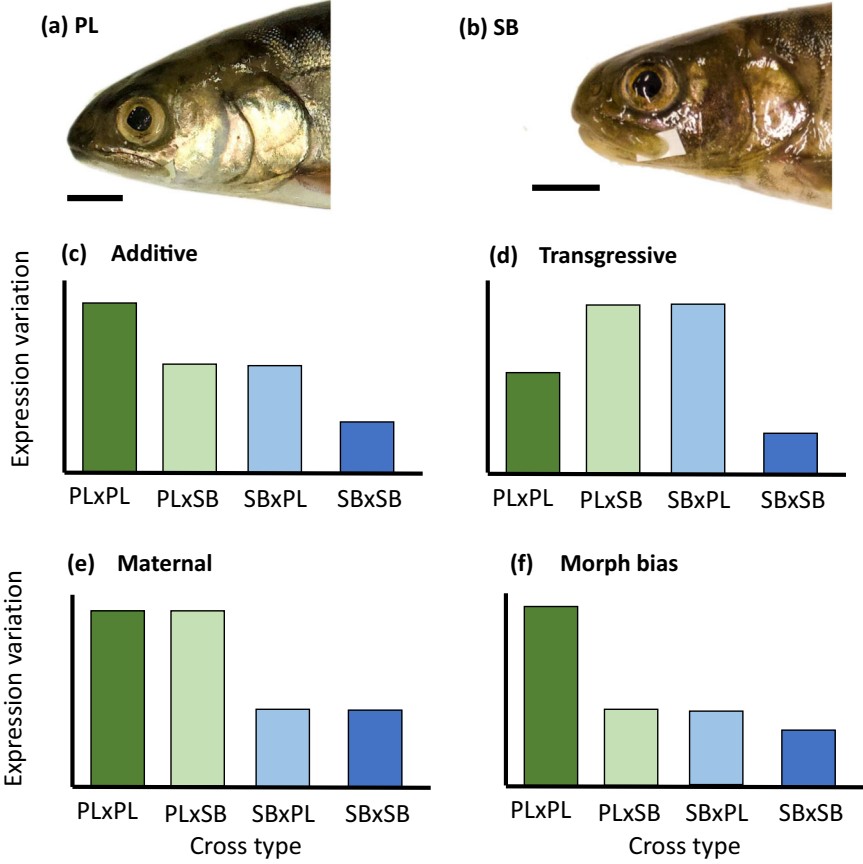

**Fig. 1 Expected patterns of gene expression variability. a**, **b** External head morphology of mature **a** Planktivorous (PL) charr, a highly specialized limnetic morph, and **b** small benthic (SB) charr, a stereotypic benthic morph. Background cropped; horizontal bar: 1 cm. **c–f** Classification of the patterns of gene expression variability among cross-types (observed as within cross-type variation in gene expression). This illustrates a hypothetical scenario with stronger canalization in the SB morph. Transgressive expression involving reduced expression variability in hybrids not shown. Cross-type: maternal morph × paternal morph. PL Planktivorous, SB small benthic.

canalization, we expected to detect an increase of gene expression variability in $F_1$ hybrids (Fig. 1c–f). We reported differences in gene expression variance scattered all over the transcriptomes of the limnetic and the benthic morphs, suggesting that mechanisms modulating gene expression enable the rapid evolution of canalization at the scale of individual traits or pathways. Furthermore, gene expression variance did not increase in hybrids unconditionally. Instead, we observed major dominance patterns, mainly maternal effects. This indicates that broad sense dominance on gene expression variability may generate developmental biases that not only facilitate divergence but also preserve the integrity of multivariate canalized phenotypes from the effects of hybridization (e.g., by maintaining the maternal patterns of expression).

## Results

**Sympatric morphs show extensive differences in expression variability.** We characterized the gene expression variability profile of each cross-type using Local Coefficients of Variation (LCVs)[54] (see "Methods"). Briefly, LCVs are unitless variability estimates based on ranking coefficients of variation for the expression of each gene, within a window of genes with similar average expression levels and assigning values ranging from (0: no variation) to 100 (maximum variation). This approach mitigates biases caused by different expression levels. Importantly, we observed similar profiles of expression variability for coding and noncoding RNAs (Figs. 2 and 3). For both mRNAs and miRNAs,

genes with covarying expression variability clustered and constituted hierarchical differences among sample groups based on developmental time points and cross-type. First, the samples clustered according to the maternal morph: the hybrids with limnetic maternal origin (PL×SB) clustered with the limnetic offspring (PL×PL), and the hybrids with benthic maternal origin (SB×PL) clustered with the benthic offspring (SB×SB). Then, the pairs of developmental time points clustered within cross-types. This indicates that not only the offspring of the two morphs have genetically based differences in gene expression variability all over the genome, but also that strong maternal effects shape this variability.

In the mRNA dataset (Fig. 2), we extracted 10 clusters of genes covarying in expression variability: (i) Clusters 2, 5, and 6 contained 4390 genes which showed a similar pattern in the hybrids as in their respective maternal morph crosses, (ii) Clusters 3 and 8 had 2719 genes with gene expression variability biased towards the limnetic morph pattern in both reciprocal hybrids, and (iii) 2445 genes from Clusters 4 and 7 showed higher variability in one hybrid cross-type than in the two pure-morph crosses (i.e., transgressive variability). For all cross-types, 2096 genes had low-expression variability (Cluster 1), and 4271 genes had high variability (Clusters 9 and 10). We investigated whether these clusters contained genes with similar functions by analyzing their associated Gene Ontology terms (GO). The genes with maternal patterns of expression variability were enriched for GO terms associated with gene regulation (Clusters 2, 5, 6) as well as head and brain development (Cluster 2, Fig. 2b and

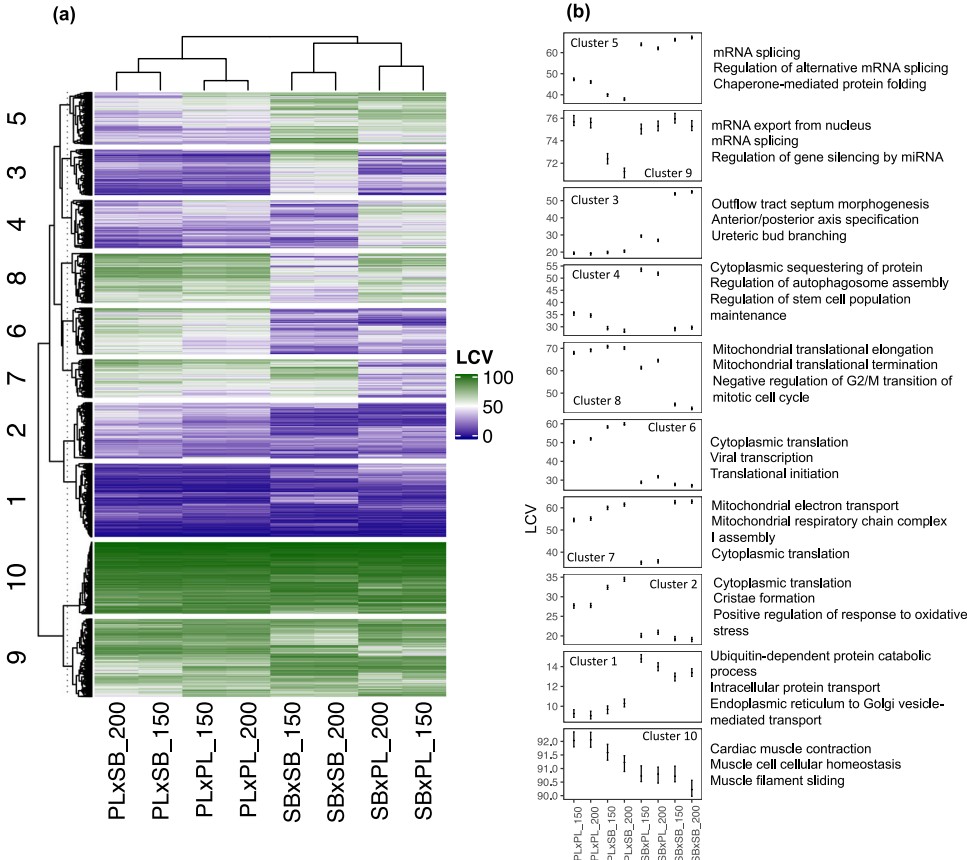

**Fig. 2 Variation in mRNAs LCV scores. a** Heatmap of LCV scores, ranging from 0 (no gene expression variation) to 100 (high gene expression variation). **b** LCV estimates (posterior modes and 95% CrIs) of the expression variability clusters and associated top-3 GO names.

Supplementary Data 1). GO terms of genes showing transgressive expression variability in the SB×PL hybrids were associated with translation, immunity and metabolism (Fig. 2b and Supplementary Data 1). Finally, genes with higher expression variability in SB×SB than in all the other cross-types were associated with muscle development, notably in the pharyngeal skeleton (Cluster 3), and genes with lower expression variability in SB×SB than in all the other cross-types were associated with gene expression regulation (Cluster 8).

The patterns of miRNAs expression variability among cross-types were less pronounced than for mRNAs, but we observed the same clustering of samples according to the maternal morph (Fig. 3a). Analyses of the 95% Credible intervals (CrIs) overlaps within the 10 gene clusters based on expression variability revealed differences between cross-types with acceptable levels of statistical certainty, even though the observed effects were fairly modest (Fig. 3b): (i) Cluster 5 showed maternally controlled expression patterns in 270 genes and Cluster 4 was associated with a similar trend (albeit with high statistical uncertainty, because of larger overlaps among 95% CrIs). (ii) Clusters 6, 8, and 9 showed 144 genes with a complex pattern with differences in expression variability between the two pure (PL×PL and SB×SB) morph crosses, and the PL×SB hybrids being similar to the PL morph while the SB×PL hybrids had intermediate variability. In all cross-types, Clusters 1 and 2 contained 293 genes with low-expression variability, Clusters 7 and 10 had 132 highly variable genes, and Cluster 3 had 164 with intermediate variability. miRNAs from all the clusters belonged to gene families expressed in various tissues, brain included (Supplementary Data 5).

We used GO enrichment analyses to investigate whether the mRNA targets of miRNAs belonging to the same cluster of expression variability were involved in specific biological processes: Various GO terms, each associated with only a few target genes of miRNAs, were observed in all clusters (Supplementary Data 2). However, we observed trends for enrichment in GO terms associated with eye and nervous system development in Cluster 2 (low-expression variability in all cross-types), with heart development in Cluster 1 (low-expression variability in all cross-types), with eye and vascular development in Cluster 5 (maternal pattern of expression), with the adrenomedullin pathway in Clusters 4 and 5 (maternal pattern of expression), and with nervous system development and epithelial cell proliferation in Clusters 8 (high variability, though significantly lower variability in the benthic morphs, SB×SB) and 10 (high variability, although trend for lower variability in SB×SB). Overall, the LCV analyses showed genetically based differences in gene expression variability during the development of the benthic and the limnetic morphs. Furthermore, maternal patterns of gene expression variability predominate in hybrids, although expression biases towards the limnetic morph or extreme variability coefficients were observed for many transcripts.

**Dominance also prevails for average gene expression.** Besides gene expression variability, we also investigated differences in average gene expression between cross-types. We observed differential expression between pure-morph crosses at both time points ($150\tau_s$ and $200\tau_s$, Fig. 4). Only 25 genes were differentially expressed at $150\tau_s$ between the benthic (SB×SB) and the limnetic (PL×PL) crosses, whereas 7824 were differentially expressed at $200\tau_s$ (adjusted $P < 0.1$). Likewise, up to 44 genes were differentially expressed at $150\tau_s$ between one hybrid cross-type and any other cross-type (SB×PL vs. SB×SB) compared with 2504 genes at $200\tau_s$ (PL×SB vs. SB×SB, Fig. 4). Thus, we only used the $200\tau_s$

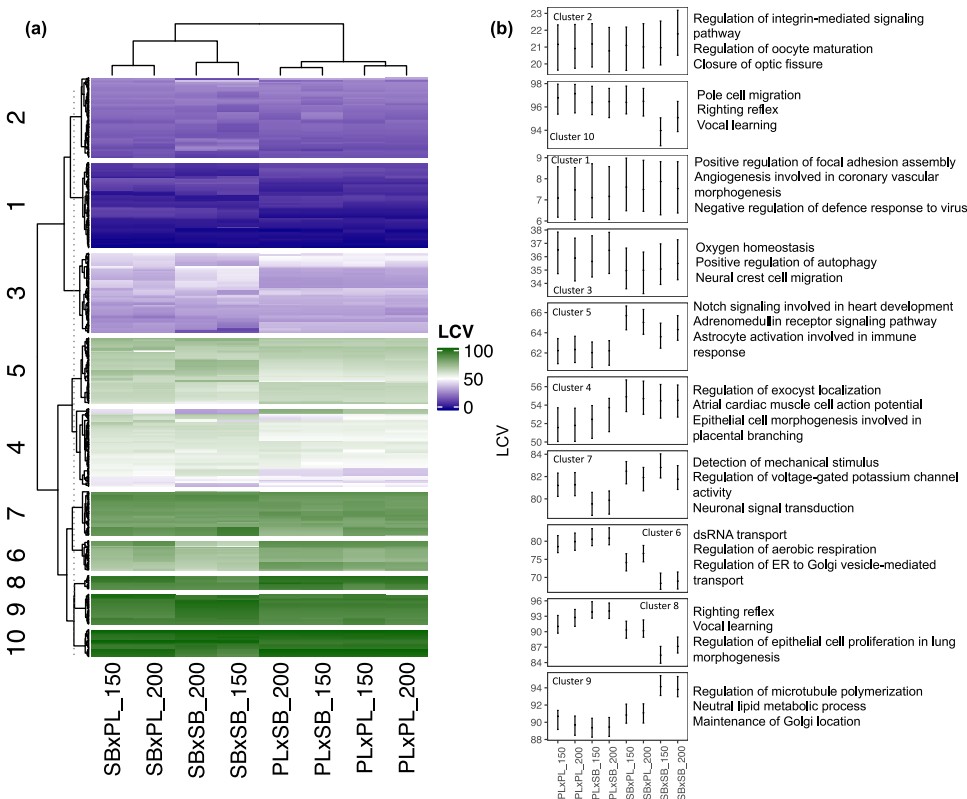

**Fig. 3 Variation in miRNAs LCV scores. a** Heatmap and **b** LCV estimates (posterior modes and 95% CrIs) of the expression variability clusters and associated top-3 GO names. LCV scores range from 0 (no gene expression variation) to 100 (high gene expression variation).

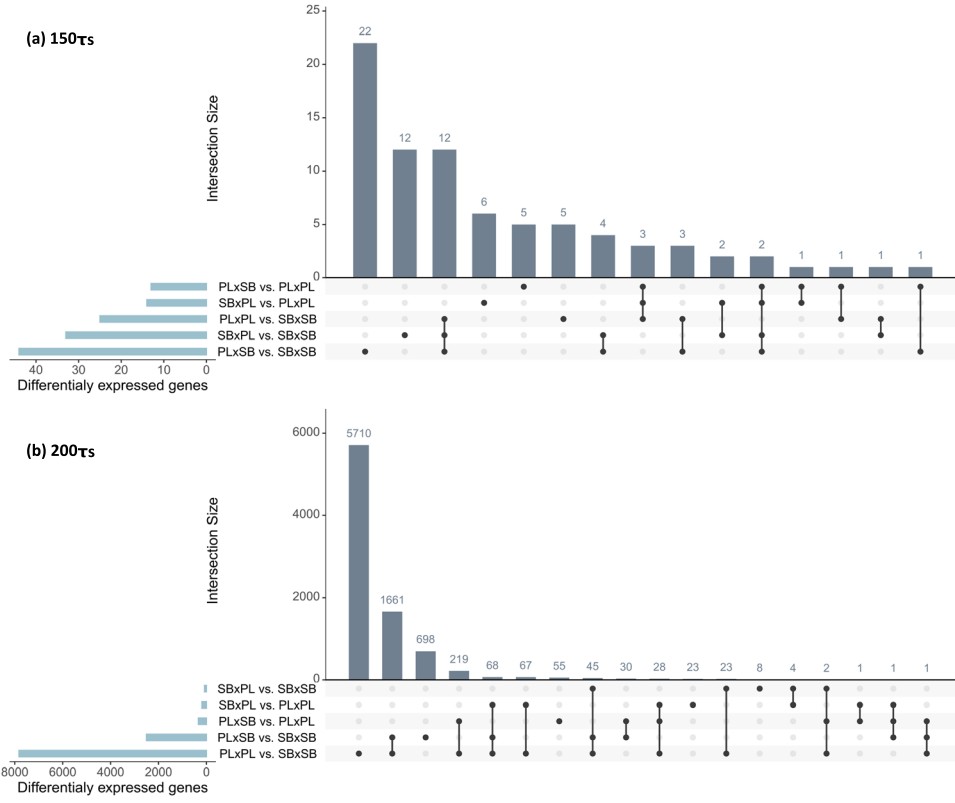

**Fig. 4 Intersections between the sets of differentially expressed mRNAs in each cross-type comparison.** Number of differentially expressed genes at **a** the early $150\tau_s$ developmental time point and **b** the latter $200\tau_s$ time point.

**Table 1 Number and proportion of differentially expressed mRNA transcripts in each contrast, $\chi^2$ and $P$ values contrast comparisons.**

| RNA | Contrast 1 | Contrast 2 | $\chi^2$ | $P$ |
|---|---|---|---|---|
| mRNA | PL×SB/PL×PL-150$\tau_s$ = 13 (0.03%) | SB×PL/PL×PL-150$\tau_s$ = 14 (0.03%) | 0.000 | 1.000 |
| | PL×SB/SB×SB-150$\tau_s$ = 44 (0.11%) | SB×PL/SB×SB-150$\tau_s$ = 33 (0.08%) | 1.299 | 0.254 |
| | PL×SB/PL×PL-200$\tau_s$ = 337 (0.82%) | SB×PL/PL×PL-200$\tau_s$ = 192 (0.46%) | 39.449 | <0.001[a] |
| | PL×SB/SB×SB-200$\tau_s$ = 2504 (6.39%) | SB×PL/SB×SB-200$\tau_s$ = 82 (0.2%) | 2339.084 | <0.001[a] |

[a]Significant after Bonferroni correction (threshold = 0.013).

time point to analyze and discuss general trends (i.e., the functional analyses and overall dominance).

At 200$\tau_s$, the majority of differentially expressed genes were unique to the benthic/limnetic contrast, suggesting intermediate expression in the hybrids (Fig. 4). Considerably fewer genes were differentially expressed between each hybrid cross-types and the cross-type of their maternal morph than with the cross-type of the alternate maternal morph. This clearly implies that maternal effects are a major a diver of average gene expression at a key developmental time point for the formation of the skull. The differentially expressed genes between the limnetic offspring (PL×PL) and either hybrid cross-types were enriched for GO terms related to metabolism, immunity and mitochondrial DNA inheritance (Supplementary Data 3). The differentially expressed genes between the benthic offspring (SB×SB) and both hybrid cross-types were enriched for GO terms associated with muscle development (Supplementary Data 3).

To further explore the extent of maternal effects in overall gene expression, we tested whether the proportion of differentially expressed mRNA genes were lower in the contrasts between pure-morph cross-types and hybrids with the same maternal morph than between pure-morph cross-types and the hybrids with a different maternal morph. Such pattern was only detected at 200$\tau_s$ in the contrasts involving the reciprocal hybrids and the SB×SB crosses (Table 1), showing that maternal effects on average gene expression are less evident when looking at the scale of the whole transcriptome.

At 150$\tau_s$, one gene was overdominant (*splicing factor U2AF subunit*), and one was under-dominant (*cytochrome c oxidase subunit*) but neither under- nor overdominance expression was detected at the 200$\tau_s$ developmental time point. While 68 genes showed a maternal pattern of expression at 200$\tau_s$, none were detected at 150$\tau_s$. We identified biased expression in hybrids towards the limnetic morph in 12 genes at 150$\tau_s$ and in 38 genes at 200$\tau_s$. The genes with maternal and limnetic-biased expression in hybrids were enriched in GO terms associated with morphogenesis in general and muscle development in particular (Supplementary Table 1). Note that the differentially expressed genes, including the ones showing maternal and limnetic-biased patterns of expression were scattered throughout the genome and not found in specific regions of differentiation (Supplementary Table 2 and Supplementary Fig. 1).

Similar results on average gene expression were observed when analyzing the miRNA data. Like in the case of mRNA expression, differential expression between pure-morph crosses was detected at both developmental time points but most of the variation appeared at 200$\tau_s$. miRNAs from 6 miRNA families differed in expression between the two pure-morph crosses at 150$\tau_s$ (Table 2). Of those, four miRNA families (miR-100, miR-181, miR-34, miR-816) contained differentially expressed genes in more than two cross-types. Two of these miRNA families included miRNAs with putative roles in brain development (miR-100 and miR-181[67,68]) and showing expression similar to the limnetic morph in both hybrid cross-types. Genes of the miR-

34 family—a family involved in brain development[69]—showed higher expression in the benthic offspring (SB×SB) compared to both hybrid cross-types, and exhibited nonsignificant but substantially large fold changes between the benthic and the limnetic cross-types. Genes of the miR-8160 family also showed higher expression in hybrids, although differences between hybrids of limnetic maternal origin (PL×SB) and the limnetic offspring were not significant. The role of miR-8160 during development is not currently known.

At 200$\tau_s$, miRNAs from 32 families were differentially expressed between the two morphs, one of those (miR-1) showing a maternal pattern of expression. Members of these families are expressed in neuronal structures, in epidermal tissues and the pharyngeal arches during zebrafish development (Table 2). Note that the predominance of these organs in our dataset may reflect literature biases. For example, there was no observed difference in the proportions of miRNAs reported to be expressed in brain tissues between our dataset of differentially expressed miRNAs and the full reference dataset of Wienholds and colleagues[67] ($X^2 = 1.82$; df = 1; $P = 0.18$).

The target genes of the differentially expressed miRNAs between pure-morph crosses were enriched in GO terms associated with numerous biological processes, including eye development and immunity at 150$\tau_s$, and eye and enteric development at 200$\tau_s$ (Supplementary Data 4). These results on *average* gene expression levels indicate substantial differential expression patterns between the benthic and the limnetic morphs, especially at the later stage of development (200 $\tau_s$). In the hybrids, maternal patterns of expression and biases toward the limnetic morph also appear to be important.

Altogether, our results show a striking consistency between gene expression variability and differential (*average*) gene expression. Specifically, we reported strong patterns of divergence in gene expression *variability* during a developmental period with considerable differential expression at the transcriptomic scale—this period corresponding to cartilage development in major structures in the head with established roles in phenotypic divergence. Taken together, our results indicate that the genes involved in such patterns of expression *variability* have putative roles in phenotypic divergence.

## Discussion

Canalization may play important roles in adaptive divergence. Any loss of buffering in the developmental mechanisms that generate phenotypic variation may expose previously cryptic variation to natural selection, while any suppression of variance may reduce the efficacy of selection[70,71]. Phenotypic variation is driven by the expression of genes that influence developmental processes, but there is little prior work on the dynamics of gene expression variance during divergence under selection. Here, we tackle this important question using sympatric Artic charr morphs. Using a common garden experimental design, we find compelling patterns of gene expression variability in embryos that depict a complex picture involving changes in many traits or

**Table 2 Differentially expressed miRNAs between pure-morph crosses, and miRNAs with nonadditive inheritance.**

| | miRNA family (homologs)[a] | Location/function of putative homologs[b] |
|---|---|---|
| DE at $150\tau_s$ | miR-100 (43), miR-148 (2), miR-181* (47), miR-199* (24), miR-375 (14), miR-455 (3) | Epithelia of pharyngeal arches, head skeleton and pectoral fins, epidermis of head, tip of the tail, brain, spinal cord thymic primordium, eyes, sense organs, in zebrafish/medaka[67,101] Cell metabolism and viability in zebrafish[102] Lens pigments epithelial cell proliferation in in newt[103] Cardiac development in zebrafish[104] Vascular development, zebrafish[105] Involved in chondrogenesis, human[106] |
| DE at $200\tau_s$ | let-7* (5), miR-1* (37), miR-10* (42), miR-124* (51), miR-125 (2), miR-128 (27), miR-132 (11), miR-138 (27), miR-148 (2), miR-181* (47), miR-193 (14), miR-199* (4), miR-20 (1), miR-200 (2), miR-203 (19), miR-206* (16), miR-2188* (5), miR-221 (1), miR-222 (20), miR-2478 (1), miR-27 (3), miR-30* (23), miR-301 (3), miR-429 (18), miR-430* (1), miR-455* (7), miR-725 (3), miR-737* (1), miR-9* (6), miR-92 (57), miR-9226 (1) | Brain, sense organs, eyes, spinal cord, skeletal muscles, gills, excretory/digestive system, pharyngeal arches, fins, epidermis of the head, tip of the tail, in zebrafish/medaka[67,101] Brain morphogenesis in zebrafish[107] Angiogenesis in muscles, in zebrafish[108] |
| Maternal at $200\tau_s$ | miR-1* (25) | Body, head, fin muscles and skeletal muscles, in zebrafish/medaka[67,101] |
| PL-dominant at $150\tau_s$ | miR-100 (43), miR-181 (47) | Brain; spinal cord, eyes, thymic primordium, sense organs, gills in zebrafish/medaka[67,101] |

[a]Different miRNAs from the same family, the same miRNAs with different orientations, paralogs or putative orthologs (the reads have aligned to different sequences, from different species, but have the same miRNAs name).
[b]Putative location/functions according to the literature.
*Also found to be differentially expressed between SB and domesticated charr embryos by Kapralova et al.[109,100].

developmental pathways, potentially leading to the multifarious differentiation of canalized phenotypes during adaptive divergence. This view is supported by the multiple clusters of genes covarying in expression variability and differing among cross-types, remarkably exceeding the expression variation attributed to developmental timing. Most of these clusters showed maternal biases in expression variability, while others showed similar expression variability in hybrids as in the limnetic morph, and some exhibited over- and underdominance in hybrids. This trend for nonadditive expression variability inheritance was substantiated by the consistency of expression profiles from two datasets involving coding and noncoding RNAs. Such results challenge our current understanding of the evolutionary-developmental dynamics of canalization by suggesting (1) the existence of unknown molecular mechanisms shaping the development of phenotypic variability and (2) the recurrence of maternal effects and dominance on those mechanisms, most likely influencing evolutionary trajectories during divergence and biasing the effects of hybridization. These two aspects will be dealt with separately in the present discussion.

First, the sharp and multiple differences in expression variance between the two morphs indicated that divergence in gene expression variability evolves rapidly and may occur multiple times across the whole transcriptome. This high evolutionary potential of gene expression variability, as suggested by the strong differences between the two recently diverging morphs, contrasts with current views on the developmental origins of phenotypic variation. Little is known about the developmental mechanisms involved in the evolution of phenotypic variability, but non-linearity in genotype–phenotype maps has been proposed as a parsimonious explanation[14] which was recently supported by experimental studies on single genes. For example, enhanced phenotypic variation can result from decelerating gene expression dose–responses curves, thereby producing the most distinct phenotypes for the same gene expression difference at the lowest gene expression levels, as observed in mice with the effects of *Fgf8* on midfacial shape[29], or with the effects of *Wnt9b* on mouth clefting[28]. Yet, the transcriptome scale snapshots provided by our

study suggest the existence of important additional mechanisms modulating canalization. In our study, those mechanisms take the form of direct changes in gene expression variability (i.e., rather than being effective downstream of the transcription) and appear to affect a multitude of genes.

The developmental implications of such changes in gene expression variability may be manyfold, but important insights can be gained through conceptual models. Waddington's epigenetic landscape, which depicts the funneling of developmental processes into valleys whose steepness represents resistance to developmental variation[13,14], is an especially powerful metaphor to envision the role of gene expression in canalization. In a context of adaptive divergence, stochastic developmental processes acting within individuals can drift towards distinct coordinates of developmental space, which ultimately correspond to phenotypes approximating contrasting fitness optima. Gene expression variability can be conceptualized as the potential energy affecting the trajectory of these developmental processes across the epigenetic landscape (i.e., the steepness of valleys[72,73]). If hybridization increases gene expression variability (as observed in hybrids between *Coregonus clupeaformis* incipient species, for example, ref. [74]), such metaphoric landscapes would flatten, resulting in wide developmental opportunities with potentially diverse evolutionary consequences (e.g., maladapted phenotypes, increased phenotypic novelty, or high resilience to incompatibilities). However, we showed that increased expression variability is not a systematic outcome of hybridization, at least regarding first-generation hybrids. Rather, maternal effects and morph biases predominate, suggesting a state of canalization in hybrids that can be conceptualized as a composite picture of multi-layered landscapes, most of those tending towards the values observed in one morph.

The consequences of this multivariate landscape of gene expression variability on the final phenotype, especially with regards to the effects of hybridization, remain unexplored. On the one hand, changes in the expression variability of genes from independent pathways may generate hybrid trait mismatches analogous to what is commonly described for average trait values,

thereby reducing fitness in hybrids. On the other hand, phenotypic robustness may be modulated by the interconnectivity of developmental pathways (e.g., from gene networks, developmental constraints, or tissue interactions). In this model, phenotypic effects resulting from the disruption of the room of maneuver of a developmental pathway (due to mutations or genetic breakdowns through hybridization) could be buffered—or accentuated—by the state of canalization of other co-acting pathways. Applying a gene regulatory network perspective[75] to variability may therefore provide efficient tools to ascertain how gene expression influences the evolution of canalization and its implications for adaptive divergence.

The other compelling aspect of our results is that, while striking evidence for the rapid evolution of gene expression variability among diverging populations was uncovered, much of this observed variation was dominated by maternal effects. Maternal effects are increasingly recognized as powerful drivers of developmental biases, propelling adaptive evolution towards specific evolutionary trajectories[7,76], and to which the Arctic charr appear to be no exception[6,77]. While the evolutionary consequences of these observed patterns remain to be established, such maternal effects might facilitate adaptive divergence by maintaining particular levels of expression variability in specific genes, and this independently of the average expression levels (i.e., regardless of the nonlinear gene expression dose–responses curves modulating variance discussed above). This would enable the emergence of multiple phenotypic differences over very short evolutionary timescales, such as the complex benthic and limnetic ecotypes in the Arctic charr of Thingvallavatn.

Early maternal effects on gene expression variability as observed in our results may not only enable the precise developmental trajectories for a given niche, but also preserve the integrity of the resulting phenotypes when populations diverge in sympatry, constraining the effects of hybridization. Recently, much attention was brought towards the role of hybridization in increasing phenotypic variation and relaxing trait covariation[78,79]. However, our results suggest that increased (gene expression) variability in hybrids may not be a dictating rule. To the contrary, we showed the predominance of maternal effects, morph biases and, to a lesser extent, transgressive variability that would result in a definite combination of trait variability. This may not only constrain the release of phenotypic variability in hybrids (so the ability to colonize new niches) but also generate reproductive isolation through lower hybrid fitness, if other dominance effects than maternal biases also act on hybrids, preventing the expression of the precise patterns of trait variability observed in either maternal lineage (i.e., mismatching trait variability). Such mismatching expression variability may especially be detrimental if hybrids are also differing from both parental lineages in average expression levels, preventing the recovery of optimal values through plastic changes (as suboptimal trait values may the locked by low-expression variability). This could be expected in our study system, since for most differential expressed genes between the planktivorous and the small benthic embryos, hybrids appeared to have average expression values laying somewhere between the two morphs (intermediate, slightly maternal or morph biased, Fig. 4).

Caution should be taken as fitness estimates related to the expression of genes identified in our study are not available. However, information on average gene expression can give further indications about the condition of F1 hybrids. Altogether, in our observations of average gene expression, the predominance of maternal biases at the scale of the transcriptome supports the view of trait mismatches. This is further supported by the identification of limnetic and maternally biased expression patterns of miRNAs with putative roles in the nervous system (miR-100,

miR-181) and muscle (miR-1) development. Therefore, hybrids exhibiting phenotypic values that are closer to one morph (and eventually present some transgressive characters) might not perform as well as the parental morphs in either of their respective niche. Surely, the developmental consequences of our gene expression estimates remain to be ascertained, notwithstanding the artificial setup of our common garden conditions, where plastic response in gene expression might have been hindered. Yet, wild small benthic and planktivorous charr are believed to utilize the same nursing grounds[60,80], most likely subjecting their embryos to a common set of environmental cues. Thus, the results from our common garden experiment enable to reasonably state that the two morphs have evolved differences in average levels and in variability of gene expression, and that the deviant values observed in hybrids for both aspects of transcription may contribute to reproductive isolation.

Finally, one may not be able to draw conclusions about the effects of hybridization on phenotypic variability without information on later-generation hybrids. More insight can be gained from the average gene expression patterns reported in hybrids between incipient species of another salmonid: the dwarf and normal whitefish, *Coregonus clupeaformis*. In this system, F1 hybrids gene expression mostly resembled the normal whitefish, and some genes were transgressive (which is comparable to our results), but transgressive expression prevailed in backcrosses[74]. Therefore, more extreme characters may be expected in later Arctic charr hybrid generations. However, the effects of hybridization on the hybrid phenotype also depend on many factors, like the genetic architecture and the selective regime[81]. Similarly, underdominance in gene expression predominate in F1 hybrids of brook charr, *Salvelinus fontinalis*[82], which differ from our results and from the findings of Renaut and colleagues[74]. Overall, the consistent patterns of nonadditive inheritance of both average gene expression and gene expression variability suggest that postzygotic reproductive isolation might already emerge for F1 hybrids of planktivorous and small benthic charr.

Regarding general evolutionary trends, our observations imply that gene expression variability, although highly evolvable itself, is under maternal effects and dominance, probably facilitating evolution in certain directions of the phenotypic space while constraining other changes. Moreover, the same processes may facilitate the maintenance of the resulting phenotypic novelty by limiting the effects of hybridization. The proximate mechanisms for this phenomenon await further studies, but their consequences most likely influence the directions of diverging evolutionary trajectories while facilitating reproductive isolation.

## Methods

**Sampling.** We collected mature small benthic (SB) and planktivorous (PL) charr in Lake Thingvallavatn with gillnets. The crossing design is depicted in Fig. 5. We generated 12 families, including 6 families of pure crosses of the benthic and the limnetic morphs (3 SB×SB; 3 PL×PL) and 6 families of reciprocal hybrids (maternal × paternal morph: 3 PL×SB; 3 SB×PL). We sampled a total of 6 embryos per family over two developmental time points (3 embryos at $150\tau_s$ and at 3 embryos $200\tau_s$). This produced a total of 72 biological replicates, so 9 embryos per cross-type at each time point. $150\tau_s$ is the time point at the onset of the pharyngeal skeleton formation, when the Meckel's cartilage (precursor of the lower jaw) and the palatoquadrate (part of the upper jaw) start appearing[66]. $200\tau_s$ corresponds to a stage when most of the cartilage structures of the head are formed and some elements (mainly the maxilla and the dentary) start ossifying[66]. The eggs were reared at ~5 °C in a hatching tray (EWOS, Norway) under constant water flow and in complete darkness at the

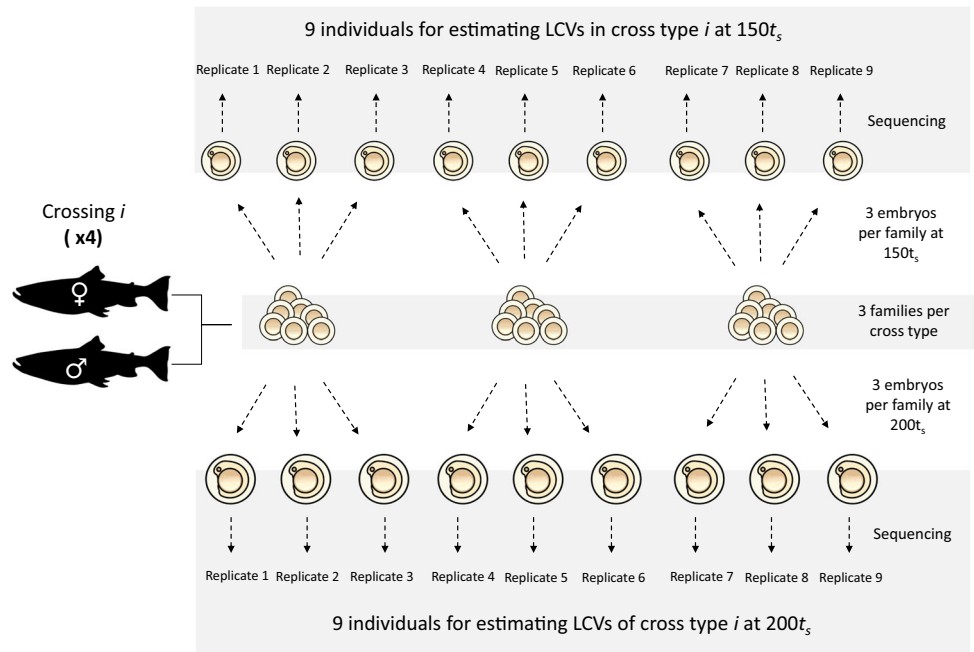

**Fig. 5 Sampling design for LCV analyses.** Embryos from gamete crosses between wild-caught individuals were reared in common garden conditions and sampled at two developmental time points: 150 and 200 tau-somite ($t_s$). Each cross-type by developmental time point category contained nine embryos from three pooled families. Icons depicting eggs and embryos are from TogoTV (DBCLS TogoTV, CC-BY-4.0 https://creativecommons.org/licenses/by/4.0/); adult charr icons are from X. Giroux-Bougard, phylopic.org (Public Domain Mark 1.0).

Holar University experimental facilities in Verið, Sauðárkrókur. Water temperature was recorded twice a day to estimate the relative age of the embryos in tau-somite units ($\tau_s$), defined as the time to form one somite pair at a given temperature[83]. Samples were flash-frozen in RNAlater (Ambion) and stored at −80 °C.

**mRNA sequencing.** Total RNA was extracted using a standard Trisol protocol (sample quality: RIN 8.5-10). The samples were sent to BGI Europe (Copenhagen, Denmark) for mRNA enrichment, purification, fragmentation, adaptor ligation, PCR and sequencing on a DNBSEQ platform.

**Small RNA sequencing.** Total RNA from the same samples as for the mRNA analyses was used for small RNA sequencing. The purity and amount of RNA was verified on a BioAnalyzer (Agilent Technologies). The samples were distributed into three batches to produce the sequencing libraries, and were prepared for sequencing following the small RNA v1.5 sample preparation protocol from Illumina. Briefly, 3' and 5' RNA adapters were ligated to small RNAs, which were subsequently, reverse transcribed into DNA and PCR amplified. The samples were then run on polyacrylamide gels and the DNA eluted from bands corresponding to 20-30 nucleotide RNA fragments. Small RNA sequencing was performed on a MiSeq platform using the MiSeq Reagent (v3) kit (Illumina).

**Data preprocessing.** The sequencing data were pre-processed following the guidelines described by Delhomme and colleagues[84]. For mRNAs, adapter removal and filtering were done by the sequencing third party, but we re-assessed reads quality with FASTQC (https://www.bioinformatics.babraham.ac.uk/projects/fastqc/). The reads were then aligned to the *Salvelinus sp.* genome[85] using STAR ([86]; settings --outSAMstrandField intronMotif --twopassMode Basic) and were counted at the gene level using FeatureCounts ([86]; settings -p -B -C). Approximately

84% of the reads mapped to unique locations of the genome (see *Data availability* section for sample-specific alignment details)[87].

For the miRNAs, we checked the quality of reads with FASTQC before and after removing the adapter sequences with Cutadapt[88]. The miRNA transcripts were then quantified with MiRDeep2[89] by preprocessing and mapping the reads to the *Salvelinus sp.* genome[85] with the Mapping module before counting the number of precursor and mature sequences with the Quantifier module, using the MiRBase reference database for all species[90]. In total, 99.94% of the processed reads from the known mature miRNA database aligned at least once with the genome (processed reads: 48,885; reported alignments: 855,281). 15.46% of the reads from our sequenced dataset aligned at least once (processed collapsed reads from all samples: 2,965,462; reported alignments: 17,099,955). Because redundant miRNA homologs can be found across species, we removed sequences exhibiting at least 95% similarity across species using Cd-hit[91]. Small RNA sequencing failed for one sample (individual from a PL×SB family at 200$\tau_s$), so we discarded it for the miRNA data analyses.

**Statistics and reproducibility.** The analyses were conducted on a set of 72 biological replicates categorized according to the sampling design described in Fig. 1. Statistical pooling of families into cross-types as described in Fig. 1 was done only for one type of analysis that did not allow considering nested designs (LCV estimates). Statistical tests, decisions, and tools used for each step of this study are explained separately in the following paragraphs.

Gene expression variability: Estimating gene expression variability is not straightforward because gene expression variance is dependent on the expression level. We applied the methods from ref. [54] to calculate Local Coefficients of Variation (LCVs) at the gene level. Briefly, an algorithm uses a sliding window on genes ordered by expression level (regardless of their location on the genome), ranks the Coefficient of Variation of the focal gene to that of the other genes located in the current

window, and determines the percentile that fits the ranking of this coefficient of variation. Hence, LCV scores are unitless estimates ranging from 0 (least variable genes) to 100 (most variable genes). We set the window size to 500 genes.

We assessed covariation in LCV estimates among cross-types with unsupervised (Kmeans) clustering[92]. Then, we extracted clusters of genes with covarying LCV with ComplexHeatmap[93]. Our decision on the number of clusters to extract (10 in both datasets) was based on the grouping of dendrogram tips. In addition, we verified the optimality of this number with the silhouette method (Supplementary Fig. 2).

LCV scores were used as a response variable in linear models (one model per cluster) to estimate gene variability differences between cross-types and the underlying inheritance pattern of the genes constituting each cluster. We fitted the models using MCMCglmm[94], specifying weakly informative priors (V = 1, nu = 0.002) and determining the quality of the output from trace plots and posterior density plots. We set the number of iterations, thinning interval, and burnin to 13,000, 10, and 3000, respectively, for mRNAs, and to 130,000, 1000, and 3000 for miRNAs. Inferences were made based on the posterior modes and the overlaps in 95% Credible Intervals.

Average gene expression: We used DESeq2[95] to estimate differential expression between cross-types for both mRNAs and miRNAs. Genes with less than 10 reads were filtered out for the downstream analyses. We corrected for false discovery by applying $\log_2$ fold change shrinkage with the `ashr` function from Stephens[96]. We included the sequencing batches (miRNA dataset) as fixed effect in the linear models. Overall dominance in gene expression was estimated by testing for differences in the proportion of differentially expressed genes between reciprocal hybrids and pure-morph crosses. We then identified candidate genes with putative dominance in expression with handwritten *R* functions, according to the rationale described in Supplementary Table 3.

Functional analyses: We inferred the mRNA targets of candidate miRNAs with miRanda[97]. Predictions were made using the mature miRNA sequences and the 3′ Untranslated Region (UTR) of the mRNA transcripts with more than ten reads in our count datasets. The 3′ UTRs were retrieved from the *Salvelinus sp.* genome[85]. We ran miRanda with default parameters and filtered the output by keeping the ten targets of each miRNA with the highest total score. miRanda is simple and efficient tool for miRNA target prediction relying on both sequence complementary and binding energy, but present the same flaws as other popular predicting algorithms regarding functional analyses[98]. In absence of experimental validations, we call for cautious interpretations of the results from this part of the analyses.

Finally, we conducted gene ontology (GO) analyses of genes from clusters exhibiting different expression variability, differential average expression, or being identified as target genes of miRNAs of interests. We performed the enrichment analyses with the topGO R package[99], using the `weight01` algorithm and making statistical inferences based on Fisher's exact test. The GO annotations were retrieved from the *Salvelinus sp.* genome repository[85].

**Ethics statement**. Sampling of wild specimens of *Salvelinus alpinus* was conducted by the authors with the permission of the Thingvellir National Park Commission and the owner of the Mjóanes farm. Z.O.J. and K.H.K. hold special permits, from the Icelandic Directorate of Fisheries, for sampling fish for scientific purposes according to Icelandic law (clause 26 of law 61/2006 on salmonid fishing). After being stripped for gametes, parent fish were killed by a sharp blow to the head and checked for absence of breathing when placed in water. Setting up crosses and the subsequent killing of parents was performed by the authors. Ethics committee approval is not needed for scientific fishing in Iceland (The Icelandic law on animal protection, Law 15/1994, last updated with Law 157/2012). Rearing of embryos was performed according to Icelandic regulations (license granted to Hólar University College aquaculture and experimental facilities). The sampling of embryos was performed by KHK. HUC-ARC has an operational license according to Icelandic law on aquaculture (Law 71/2008), that includes clauses of best practices for animal care and experiments. All the individuals sequenced for this study were at a pre-hatching stage, before noticeable sex differentiation.

**Reporting summary**. Further information on research design is available in the Nature Portfolio Reporting Summary linked to this article.

## Data availability
Raw sequencing data and processed reads were deposited on Gene Expression Omnibus and are publicly available under the accession GSE193797. Source data and metadata for Figs. 2–4 are accessible in Supplementary Data 6. All other data are available at https://github.com/quentin-evo/rna-charr.

## Code availability
R codes[100] and preprocessing log files are available at https://github.com/quentin-evo/rna-charr.

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

## Acknowledgements

We thank Sigurður S. Snorrason, Skúli Skúlason, Neil Metcalfe, Michael Morrissey, and Arnar Pálsson for their theoretical guidance. We also thank Oren Raz, Sébastien L.T. Matlosz, Lea J. Plesec, Marina de la Cámara, Jóhannes Guðbrandsson, and Ingeborg J. Klarenberg for their help in processing and analyzing the data. We are grateful to Bjarni K. Kristjánsson, Kári H. Árnason, Rakel Þorbjörnsdóttir, and Christian Beuvard for the organization and the maintenance of the experimental setup. We thank Baldur Kristjánsson for his help with Bioanalyzer and Miseq. Fieldwork was conducted with help from the Arctic Charr and Salmonids Group members (University of Iceland) and the farmers of Mjóanes, Jóhann Jónsson and Rósa Jónsdóttir. This work was founded by the Icelandic Centre of Research, Rannís (Postdoctoral Grant 152406051, Icelandic Research Fund Grant 173802051, Doctoral Student Grant 6001692039).

## Author contributions

Conceived the experiments: K.H.K., Z.O.J., and Q.J.H. Data collection: K.H.K., Z.O.J., D.A.T., and Q.J.H. Molecular work: K.H.K. Data analyses: Q.J.H. and D.A.T. Manuscript writing: Q.J.H., K.H.K., Z.O.J., D.A.T., and B.H.

## Competing interests

The authors declare no competing interests.
