## [Peer Review File · Communications Biology]

Reviewers' comments:

Reviewer #1 (Remarks to the Author):

It was a pleasure to read this nice study on gene expression levels and variability in crosses of Arctic charr morphs by Horta-Lacueva et al. This is an interesting addition to several research directions: hybridization (especially in fish), evolution of gene expression and the role of variability, and the evolution of canalization. The study is overall well conducted, and the authors present clear and testable predictions in the introduction.

My main comment is that variability is computed on only 3 replicates per condition, which is very little. How can the authors convincingly show that these estimates are robust?

Other detailed comments:

p. 4, the authors write that they "observed similar profiles of expression variability for both coding and non-coding RNAs", but the coding mRNA profiles are characterized by large differences in variability patterns between crosses, whereas for the miRNAs there are hardly any such differences. This should be clarified.

p. 4 "the overall level of gene expression variability (i.e., for all RNAs in each dataset) was similar among cross types": if I understand well the method used, as it is based on ranks, the overall expression variability will always be similar in each sample by definition.

Discussion: please note that the common garden experiment limits the study of canalization to genetic differences; there could be gene x environment interactions especially in variability which would not be visible here, but would affect the conclusions on canalization.

p. 10, please restate the meaning of the abbreviations SB and PL at the start of the Materials and Methods.

Please make the "handwritten R functions" available as supplementary materials.

p. 13, miRanda is quite an old method, and was established for model organisms such as *Drosophila melanogaster*. The authors should justify why they use this method, and how robust it is to non model species genomics.

Some benchmark / review papers as starting points:

<https://academic.oup.com/bib/article/16/5/780/217102>

<https://academic.oup.com/bib/article/21/6/1999/5618833>

Table 1: please give exact p-values, as well as the Bonferroni threshold used.

Table S4: I checked a few genes, and there isn't always a clear correspondance to *Danio rerio* genes.

For example the first row is "miR-30", but in *Danio rerio* I find 5 mir-20 genes:

<https://www.mirbase.org/summary.shtml?fam=MIPF0000005>

Reviewer #2 (Remarks to the Author):

Paper review

Manuscript: Evolution of canalization: Lessons from a classic case of resource polymorphism

General comments:

1. There are quite a bit of spelling, formatting, and grammatical mistakes within the manuscript and it would be great to see it cleaned up. Another pass at editing would greatly improve the manuscript.
2. The results section has been written very technically and has a lot of information in it. It would be great if the authors could place the various expression results in a biological context and how they may relate to adaptive divergence. This does come up in the discussion but, it would be nice for the readers to be able to take away the key points of the various results quickly and succinctly without a lot of the technical jargon associated with them.
3. Currently the discussion is a bit jumbled with ideas, especially the first paragraph. The authors need to A) place their results in the biological context of adaptive divergence and B) follow the outline they used in the introduction, i.e. focus on canalization of phenotypic variation, then dominance, then maternal effects.
4. It would make the manuscript a bit more clear if the authors could further differentiate their tests of dominance and maternal effects throughout the manuscript. Currently, it's tough to see how they are different within these tests as dominance of traits relates heavily within maternal effects.
5. The discussion of the manuscript needs a lot of work. Currently the discussion is focused on a lot of general ideas that should be present (but are not) within the introduction. I would like to see how your specific results support the canalization of phenotypic variation between the benthic and pelagic morphs. I would also like to see how this directly compares to previous studies within this system that support canalization of phenotypic variation (see Parsons et al., 2010 & 2011).
6. The methods section seems pretty solid but could use some fine tuning in language. I would like to see the methods section be a bit more clear and relate back to the biological significance of each test. Why and how do these tests relate to specific biological phenomenon (re: canalization, dominance, and maternal effects). Be specific and avoid heavily technical jargon.
7. Overall, I like this manuscript and think it sheds light on some interesting ideas relating to adaptive divergence within Icelandic Arctic charr. However, I would like to see the manuscript be a bit more clear and show specifically how their results advance the theory of adaptive divergence in sympatry.

Specific comments:

Line 21. Many developmental processes influence an organisms response to natural selection.

Lines 22-23. Not a big fan of this definition of canalization as it is very vague. Canalization occurs when phenotypic variation is reduced throughout ontogeny while environmental and genetic variance remains relatively constant.

Line 23. Canalization not canalisation.

Line 26. Remove dominance as it is poorly characterized in this sentence. This might help streamline a couple ideas in the abstract.

Line 47. Please provide a better definition of canalization from a phenotypic perspective as the subsequent sentence is highly vague and doesn't entirely make sense.

Lines 48 and 49. One could argue that we have a decent picture of how populations evolve during sympatric speciation. Please focus this on the evolvability of molecular pathways that lead to canalized phenotypes. That's the main point of this manuscript.

Lines 54-64. I agree with the main message of this paragraph and attempting to assess the contribution of trait dominance and parental effects are important in adaptive divergence research.

However, this paragraph would greatly benefit from a streamlining the ideas within it.

Lines 66-67. Why are gene expression studies well suited for assessing phenotypic canalization?

Line 67. Knowledge on variation in gene expression modules...

Line 70. What techniques and why are they important to assessing canalization? Are these techniques not used in studies assessing adaptive divergence? MicroRNA techniques have been used before.

Line 79-81. Why did you focus on these two morphs in particular? There are four morphs within that lake and three of which are highly genetically divergent. Why was the large benthic morph not included in the manuscript? It makes sense (to a degree) to not include the large benthic morph as it spawns much earlier than the others which makes making crosses tricky but, crosses between the large benthic and pelagic morphs have been made before (see Parsons et al., 2010 & 2011). This should be stated in the manuscript when describing the study system.

Line 91. I would like to see a better topic sentence that introduces the overall goal of the study.

Line 99. There is a labeling issue with figure 1. In the actual figure, Figure 1b occurs twice. In the text they mention Fig. 1c-f which does not match the actual figure.

Line 108. Does a value of 100 LCV mean there is a 100% variability? What is the scale that this coefficient is working on? What does the difference between 0 and 100 mean?

Line 119. Is there was way to make the numbered clusters more clear in figure2? It's hard to tell where on the hierarchical plot they line up. Maybe put the cluster numbers on the other side of the plot.

Line 160. Cite figure 4 here as the data within this figure is relevant to the text.

Line 235. The topic paragraph of the discussion should have an overview of how the main results from this manuscript provide novel information in the context of adaptive divergence. Currently, this is a bit lost in the first paragraph of the discussion. Focus this first on how the results support or do not support canalization of phenotypic variation between the two morphs. Then focus on dominance and maternal effects as this is how the introduction was laid out.

Line 251-262. First how does your results support the canalization of the phenotypes of the benthic and pelagic morphs? Then how does that relate to the literature? It's currently hard to see how your results are novel within the context of adaptive divergence. Place your results within the larger context and show why they are novel.

Lines 264-289. This is great information but should definitely be summarized and brought up in the introduction. Then use the scaffold of the introduction to show how your results support or do not support these ideas. Currently, this is not really a discussion of how your work supports these ideas but just a summarization of previously established ideas.

Line 550. I would cite Brachmann et al., 2022. Variation in the genomic basis of parallel phenotypic and ecological divergence in benthic and pelagic morphs of Icelandic Arctic charr (*Salvelinus alpinus*). It may provide context in which the need to assessing the development of these morphs may fit.

Response to referees

Reviewer #1 (Remarks to the Author):

It was a pleasure to read this nice study on gene expression levels and variability in crosses of Arctic charr morphs by Horta-Lacueva et al. This is an interesting addition to several research directions: hybridization (especially in fish), evolution of gene expression and the role of variability, and the evolution of canalization. The study is overall well conducted, and the authors present clear and testable predictions in the introduction.

My main comment is that variability is computed on only 3 replicates per condition, which is very little. How can the authors convincingly show that these estimates are robust?

We appreciate Reviewer 1's positive comment on the scientific relevance of our study and on the quality of our work. Reviewer 1's main concern was about our sample size. We acknowledge that the description of our experimental design lacked clarity, which we improved accordingly (line 462-472). For further readability, we also added a table describing the sampling design (Table 3). We are confident in that it is now clear that we used 9 replicates per condition for the LCV analyses, specifically (3 pooled families with 3 sampled embryos in each for each cross type-time point category).

We understand Reviewer 1's uncertainty about our sample size, so we have run a small simulation to demonstrate the validity of our design (Fig. R1, code below). We estimated Coefficients of Variation with sample sizes ranging from 3 to 20 individuals and for low (0.10), intermediate (0.5) and high (0.75) simulated CV values. Looking at the standard deviation of CV calculated in 100 datasets for a given sample size, we can observe that the variation in CV estimated with 9 individuals is reasonably close to what is achieved with bigger datasets. For our experiment, this indicates that having 9 individuals per category is sufficient to obtain estimates that are not much less precise than with a larger sample size. Therefore, we are confident in that our sampling design satisfies a fair trade-off between reasonable statistical requirement and realistic sequencing costs.

Figure R1. Standard deviations of 100 CV estimates calculated with sample sizes ranging from 3 to 20. The estimates were calculated from three dataset of with true LC of 0.10, 0.50 and 0.75, respectively.

Code:

```
lc_0.1<- as.data.frame(matrix(nrow = 18,ncol = 3)) # dataframe for sd of CV estimates (1000 datasets
per sample size - the sample sizes ranging from 3 to 20)
colnames(lc_0.1) <- c("n","sd.LC","true.LC")
lc_0.1$true.lc <- c("0.1")
lc_0.1$n <- c(3:20)

for(b in 1:nrow(lc_0.1)){
  var.e <- c()

  for(i in 1:100) {
    y <- rnorm(n = b,mean = 100,sd = sqrt(10)) # generate individuals with true CV = 0.1

    var.e[i] <- var(y)/mean(y) # LC estimate of individual dataset
  }
  lc_0.1$sd.LC[b] <- sd(var.e) # standard deviation of LC estimates
}

#### Same with true CV = 0.5

lc_0.5<- as.data.frame(matrix(nrow = 18,ncol = 3))
colnames(lc_0.5) <- c("n","sd.LC","true.LC")
lc_0.5$true.lc <- c("0.5")

lc_0.5$n <- c(3:20)

for(b in 1:nrow(lc_0.5)){
  var.e <- c()

  for(i in 1:100) {
    y <- rnorm(n = b,mean = 100,sd = sqrt(50))
    var.e[i] <- var(y)/mean(y)
  }
  lc_0.5$sd.LC[b] <- sd(var.e)
}

#### Same with true CV = 0.75

lc_0.75<- as.data.frame(matrix(nrow = 18,ncol = 3))
colnames(lc_0.75) <- c("n","sd.LC","true.LC")
lc_0.75$true.lc <- c("0.75")
```

```

lc_0.75$n <- c(3:20)

for(b in 1:nrow(lc_0.75)){
  var.e <- c()

  for(i in 1:100) {
    y <- rnorm(n = b,mean = 100,sd = sqrt(75))
    var.e[i] <- var(y)/mean(y)
  }
  lc_0.75$sd.LC[b] <- sd(var.e)
}

lc_m <- rbind(lc_0.1,lc_0.5,lc_0.75)

# Plot results
library(ggplot2)
ggplot(lc_m, aes(x=n, y=sd.LC, colour= true.lc)) +
  geom_line() +
  geom_point(size = 3) + theme_bw() +
  theme(panel.grid.major = element_blank(),
        panel.grid.minor = element_blank()) +
  ylab("sd (LC estimates)") +
  #scale_x_continuous(breaks = c(0,2,4,6,8,10,12,14,16)) +
  #scale_fill_manual(values=c( "#E69F00", "#56B4E9")) +
  #scale_color_manual(values=c( "#E69F00", "#56B4E9")) +
  xlab("n individuals") +
  theme(axis.text = element_text(),
        axis.text.x = element_text(size = 14),
        axis.text.y = element_text(size = 14),
        axis.title.x = element_text(size = 14),
        axis.title.y = element_text(size = 14))

```

Other detailed comments:

p. 4, the authors write that they "observed similar profiles of expression variability for both coding and non-coding RNAs", but the coding mRNA profiles are characterized by large differences in variability patterns between crosses, whereas for the miRNAs there are hardly any such differences. This should be clarified.

We agree with Reviewer 1 about the modest differences in expression variability in miRNAs. Yet, our claim is that the observed variation was sufficient to produce the same clustering among samples as in the mRNA dataset (along the horizontal dimension of the heatmaps), that is: the samples clustered by cross types rather than time points, and that the hybrids were associated with the maternal morph. It is in light of this aspect that we stated a striking similarity between the two datasets. We clarified this statement. We also emphasised the magnitude of the effects and their statistical uncertainty by adding descriptions of the confidence intervals overlaps in the results (lines 192-197).

p. 4 "the overall level of gene expression variability (i.e., for all RNAs in each dataset) was similar among cross types": if I understand well the method used, as it is based on ranks, the overall expression variability will always be similar in each sample by definition.

Review 1 is right in that this result does not convey strong biological information. We deleted this sentence and the related Supplementary Figure.

Discussion: please note that the common garden experiment limits the study of canalization to genetic differences; there could be gene x environment interactions especially in variability which would not be visible here, but would affect the conclusions on canalization.

This is indeed a good point. Yet, our study can uncover genetic effects as well as maternal effects (that are, plastic responses induced by the genitor through niche construction – in our case, this would most likely be egg yolk quantity and molecular composition). Note that we elaborated our thoughts on maternal effects in the discussion while avoiding speculations on the proximate mechanisms explaining our results (lines 384-448).

“External” environmental effects (physicochemical variables in the nursing habitat + maternal effects related to the construction of the spawning “nest”) might be interesting to consider, but in the wild, embryos of both morphs incubate within the same nursing grounds. We are not aware of small-scale environmental variables that might trigger plastic differences between morphs. Surely, different genotypes might differ in their phenotypic response to the variables of an artificial setup, which is inherent to virtually all common garden experiments. We added a statement to acknowledging aspects (lines 426-428)

p. 10, please restate the meaning of the abbreviations SB and PL at the start of the Materials and Methods.

We restated of the meaning of these abbreviations.

Please make the "handwritten R functions" available as supplementary materials.

The scripts are publicly available at <https://github.com/quentin-evo/rna-charr>. This link is now included in the data availability statement.

p. 13, miRanda is quite an old method, and was established for model organisms such as *Drosophila melanogaster*. The authors should justify why they use this method, and how robust it is to non model species genomics.

Some benchmark / review papers as starting points:

<https://academic.oup.com/bib/article/16/5/780/217102>

<https://academic.oup.com/bib/article/21/6/1999/5618833>

Reviewer 1 made two remarks on our decision predict miRNA target with miRanda, being (1) the release date of the algorithm and (2) its initial development for *Drosophila* studies. We will answer these two points separately:

1. 19 years have passed since the first release of miRanda, but age does not systematically mean obsolescence. Indeed, miRanda is still widely used by the research community, as no less than 1290 published articles have referred to the original paper in the last four years. We are aware of more recent and elaborate algorithms (e.g., accounting for evolutionary conservation as with TargetScan, enabling secondary structure modelling, etc.), but our specific query regarding this part of the analyses is fairly simple: to find the intersecting mRNAs and miRNAs from our two datasets based on 3'UTR matching. miRanda provides a robust way to achieve this with an already advanced method (not only by returning seed matching but also scoring the binding energy). Most importantly, target predicting for GO analyses is a tricky enterprise, and recent algorithms like TargetScan have been demonstrated to present the same shortcomings as miRanda for this kind of analyses (Fridrich, Hazan, & Moran, 2019).

2. The first methods paper on miRanda was indeed tested on *Drosophila*, but proved to be performant with vertebrates (human and fishes) by the same developers (<https://dx.plos.org/10.1371/journal.pbio.0020363>). Also, in our case the reference miRNAs were from our own alignments (MirDeep2 output) and the 3'UTR were extracted from the charr (*Salvelinus sp.*) genome, which solve the issue of using reference targets from another model species. Moreover, the principle of 3'UTR recognition is admitted to be highly conserved among animals (10.1073/pnas.0603529103, 10.1038/s41580-018-0045-7), so we have no reason to deem miRanda inadequate for analyses with our model.

Therefore, we are very confident the suitability of miRanda compared to other available algorithms. Reviewer 1's questioning is legit, and we added a disclaimer on this in the Methods section (lines 543-546).

Table 1: please give exact p-values, as well as the Bonferroni threshold used.

We added the exact values and the threshold for Bonferroni correction.

Table S4: I checked a few genes, and there isn't always a clear correspondence to *Danio rerio* genes. For example the first row is "miR-30", but in *Danio rerio* I find 5 mir-20 genes: <https://www.mirbase.org/summary.shtml?fam=MIPF0000005>

Indeed. We clarified this by adding the name of the putative homologs in the second column of Table S4.

Reviewer #2 (Remarks to the Author):

Paper review

Manuscript: Evolution of canalization: Lessons from a classic case of resource polymorphism

General comments:

1. There are quite a bit of spelling, formatting, and grammatical mistakes within the manuscript and it would be great to see it cleaned up. Another pass at editing would greatly improve the manuscript.

The manuscript has been significantly reshaped. We have now carefully checked the spelling for American English.

2. The results section has been written very technically and has a lot of information in it. It would be great if the authors could place the various expression results in a biological context and how they may relate to adaptive divergence. This does come up in the discussion but, it would be nice for the readers to be able to take away the key points of the various results quickly and succinctly without a lot of the technical jargon associated with them.

We thank the reviewer for the advice. We added sentences throughout this section contextualizing the output of our analyses and summarizing the logic of some analyses.

3. Currently the discussion is a bit jumbled with ideas, especially the first paragraph. The authors need to A) place their results in the biological context of adaptive divergence and B) follow the outline they used in the introduction, i.e. focus on canalization of phenotypic variation, then dominance, then maternal effects.

We acknowledge that the structure of our discussion was not explicit. We now made it clearer by dividing the discussion into subsections. We also added more information to contextualize our results in a context of adaptive evolution in general, and of sympatric speciation in particular. The discussion is now streamlined and adopts a classical narrow-

to-wide structure (from discussing the molecular patterns to relating them to evolutionary consequences), as follows:

1. A first paragraph reminding the theoretical issue and summarizing the results.
2. An Evo-Devo oriented section on the rapid changes in gene expression variability. It shows how our results compare with the current state of knowledge on the molecular processes underlying canalisation (paragraph 1) and which news information they bring on this issue (paragraphs 2 and 3).
3. A subsection on maternal effects and dominance, broadening the perspective towards eco-evolutionary processes. This subsection first interprets our results based on the current theoretical view on parental effects (paragraph 1), before discussing how they might influence the emergence of phenotypic divergence (paragraph 2) and the maintenance of such divergence through reproductive isolation (paragraph 3). This part refers to the Arctic charr of Thinnvallvatn as an informative study case, putting our results in perspective with previous studies on this system.

Given the very interpretative nature of this discussion, we ended this subsection by acknowledging the explanatory limitations of our data while referring to other empirical studies justifying to which extent our claims can be supported.

4. We added a Conclusion to wrap-up the main findings.

4. It would make the manuscript a bit more clear if they authors could further differentiate their tests of dominance and maternal effects throughout the manuscript. Currently, it's tough to see how they are different within these tests as dominance of traits relates heavily within maternal effects.

It is indeed very tricky to separate the genetic effects of dominance *stricto sensu* from parental effect. We address this aspect within the limits of our experiments by referring to such dominance mechanisms together with maternal effects under the broad definition of dominance (as in Thompson et al. 2019). There is already a disclaimer on this, earlier in the Introduction (l. 85-86). What can be differentiated between dominance and maternal effect was made explicit in Figure 1c-f.

5. The discussion of the manuscript needs a lot of work. Currently the discussion is focused on a lot of general ideas that should be present (but are not) within the introduction. I would like to see how your specific results support the canalization of phenotypic variation

between the benthic and pelagic morphs. I would also like to see how this directly compares to previous studies within this system that support canalization of phenotypic variation (see Parsons et al., 2010 & 2011).

We modified the Discussion as described in our response to Comment 3.

Note that we are well aware of Parsons et al., 2010 & 2011. However, assuming the state of canalisation of one morph relative to the other on their phenotype remains delicate in light of the current state of knowledge. This is a current point of contention among the co-authors of the present manuscript. This especially holds as many questions remain unresolved about which phenotype is more specialised than the other and about which traits are derived from common ancestors (e.g. counterintuitive results from behavioural traits in Skulason et al., 1993). Intersecting our results (molecular levels) with specific morphological (e.g. from Parsons et al., 2010 & 2011) or behaviour measurement would require several explanatory leaps. A thorough discussion on this aspect would be beyond the scope of the present paper. We therefore decided to only refer to these studies when introducing the study system in the introduction (l. 119-121 and l.132-134).

6. The methods section seems pretty solid but could use some fine tuning in language. I would like to see the methods section be a bit more clear and relate back to the biological significance of each test. Why and how do these tests relate to specific biological phenomenon (re: canalization, dominance, and maternal effects). Be specific and avoid heavily technical jargon.

We wish to keep the Methods brief and factual to correspond to the style of the journal. While the logical flow of the experiment is announced in the Results (appearing first according to the journal format), our Methods section provides a minimum of technical information for replicating each step and assessing their practical quality. We acknowledge that the technical terms can be disconcerting, but we judge them necessary to clearly communicated the main decisions taken for the sequencing parts and for conducting the bioinformatic analyses.

7. Overall, I like this manuscript and think it sheds light on some interesting ideas relating to adaptive divergence within Icelandic Arctic charr. However, I would like to see the manuscript be a bit more clear and show specifically how their results advance the theory of adaptive divergence in sympatry.

We are very pleased that Reviewer 2 appreciates the scientific value of our study. Reviewer 2's comments on how the data was presented and contextualized was fair. While we disagree with Reviewer 2 about putting an emphasis on the Arctic charr, which we used to investigate broad biological questions, we have restructured the discussion, which now relates our results to the expected evolutionary implications of canalization (that gene expression variability evolves rapidly and directs the evolutionary trajectories of sympatric populations through maternal effects).

Specific comments:

Line 21. Many developmental processes influence an organism's response to natural selection.

Corrected.

Lines 22-23. Not a big fan of this definition of canalization as it is very vague. Canalization occurs when phenotypic variation is reduced throughout ontogeny while environmental and genetic variance remains relatively constant.

We disagree with Reviewer 2's suggested definition, which is more an expected outcome of canalization rather than a definition. We used the definition provided in Hallgrímsson et al. (2002). We acknowledge that this definition may be vague, but relying on a precise, mechanistic definition of canalization can be prone to explanatory pitfalls as discussed in Hallgrímsson *et al.* (2019).

Line 23. Canalization not canalisation.

Corrected.

Line 26. Remove dominance as it is poorly characterized in this sentence. This might help streamline a couple ideas in the abstract.

Done.

Line 47. Please provide a better definition of canalization from a phenotypic perspective as the subsequent sentence is highly vague and doesn't entirely make sense.

We maintain that we judge more appropriate to keep this general definition of canalization, as explained in our response to the second specific comment.

Lines 48 and 49. One could argue that we have a decent picture of how populations evolve during sympatric speciation. Please focus this on the evolvability of molecular pathways that lead to canalized phenotypes. That's the main point of this manuscript.

We disagree with Reviewer 2's assertion that one could argue that we have a decent picture of sympatric speciation. Sympatric speciation has been constantly dismissed and redeemed (hence Mayr's metaphor of the Lernean Hydra), and few empirical cases could establish the occurrence of this phenomenon up until recently (Coyne & Orr, 2004). Only recent breakthrough of speciation genomics has ascertained the commonality of sympatric speciation while revealing unexplored aspects of the build-up of genetic differences (Nosil, 2012; Foote, 2017). Furthermore, the evolutionary processes related to the emerge of reproductive barriers in sympatry and their relative potential to counter the effects of gene flow remain a black box (Irwin, 2020). We therefore have no reason to question the validity of this statement. The effects of hybridization on the evolutionary dynamics of canalisation is a major aspect of our study, so we judge necessary to introduce the inherent issue of sympatry.

Lines 54-64. I agree with the main message of this paragraph and attempting to assess the contribution of trait dominance and parental effects are important in adaptive divergence research. However, this paragraph would greatly benefit from streamlining the ideas within it.

This part has been changed together with the entire rewriting of the introduction.

Lines 66-67. Why are gene expression studies well suited for assessing phenotypic canalization?

We elaborated on this (l. 100-108).

Line 67. Knowledge on variation in gene expression modules...

We corrected this.

Line 70. What techniques and why are they important to assessing canalization? Are these techniques not used in studies assessing adaptive divergence? MicroRNA techniques have been used before.

The novelty here is that *variability* is studied (instead of average changes), as explained in the first part of the paragraph. Further details on those techniques are provided in the references given and are described in the Methods. For the sake of the clarity and the synthetic style of the Introduction, we wish to keep the level of details as it is.

Line 79-81. Why did you focus on these two morphs in particular? There are four morphs within that lake and three of which are highly genetically divergent. Why was the large benthic morph not included in the manuscript? It makes sense (to a degree) to not include the large benthic morph as it spawns much earlier than the others which makes making crosses tricky but, crosses between the large benthic and pelagic morphs have been made before (see Parsons et al., 2010 & 2011). This should be stated in the manuscript when describing the study system.

SB and PL charr are the only two morphs that both represent genetic differentiated populations and show major spatiotemporal overlap when spawning. As Reviewer 2 acknowledged, the large benthic (LB) charr are strongly reproductively isolated from the other morphs by spawning much earlier than SB and PL charr and by being restricted to particular spawning grounds (Skúlason *et al.*, 1989; Horta-Lacueva *et al.*, 2022). While investigating the evo-devo origins of LB charr would definitely provide interesting insights on this particular study system, it would be beyond the scope of the present study.

Note that we did not find reference to hybrids between LB and PL in Parsons et al., 2010 & 2011. To our experience, obtaining reasonably a large number of families of reciprocal hybrid crosses involving LB charr and rearing the embryos simultaneously with PL and SB cross would – besides complicating the study – requires solving major logistic and technical burdens of (e.g., gamete freezing).

Therefore, we do not judge necessary to discuss the omission of an aspect that is both hardly applicable and without which our study can stand alone.

Line 91. I would like to see a better topic sentence that introduces the overall goal of the study.

This sentence has been removed as that the introduction was re-written.

Line 99. There is a labeling issue with figure 1. In the actual figure, Figure 1b occurs twice. In the text they mention Fig. 1c-f which does not match the actual figure.

We fixed Fig. 1 as shown below, and we correct for the references to it in the text (l. 141):

Line 108. Does a value of 100 LCV mean there is a 100% variability? What is the scale that this coefficient is working on? What does the difference between 0 and 100 mean?

We added the information that LCV are unitless estimates (lines 156-160). We also added a reference to this in the Methods section, where the logic behind the algorithm is explained (lines 508-515).

Line 119. Is there was way to make the numbered clusters more clear in figure2? It's hard to tell where on the hierarchical plot they line up. Maybe put the cluster numbers on the other side of the plot.

We acknowledge that the arrangement of Figure 2 is complex. However, this is the best compromise we obtained between readability, the minimum of information to be conveyed and programming limitations.

Line 160. Cite figure 4 here as the data within this figure is relevant to the text.

We added a reference to Fig. 4.

Line 235. The topic paragraph of the discussion should have an overview of how the main results from this manuscript provide novel information in the context of adaptive divergence. Currently, this is a bit lost in the first paragraph of the discussion. Focus this first on how the results support or do not support canalization of phenotypic variation between the two morphs. Then focus on dominance and maternal effects as this is how the introduction was laid out.

The discussion has been reshaped, including parts of the first paragraph, as described in our response to the general comment number 3.

Line 251-262. First how does your results support the canalization of the phenotypes of the benthic and pelagic morphs? Then how does that relate to the literature? It's currently hard to see how your results are novel within the context of adaptive divergence. Place your results within the larger context and show why they are novel.

As written above, this part has been considerably rewritten. We developed this section and discussed the evolutionary implication of our results (lines 334-448).

Lines 264-289. This is great information but should definitely be summarized and brought up in the introduction. Then use the scaffold of the introduction to show how your results support or do not support these ideas. Currently, this is not really a

discussion of how your work supports these ideas but just a summarization of previously established ideas.

We also reshaped this part and have rewritten the introduction. Our discussion relates to the questions enounced in the introduction (1. that canalisation rapidly evolves, 2. that this evolution is biased, thereby affecting the effects of hybridization). Unlike Reviewer 2, we don't think the ideas enounced in this part of the discussion are well established. We introduced these concepts to show that our results actually imply additional mechanisms. Some of these ideas put our results into perspective but cannot be tested directly with the data in hand, making them appropriate for a Discussion but too expedite to be in the Introduction.

Line 550. I would cite Brachmann et al., 2022. Variation in the genomic basis of parallel phenotypic and ecological divergence in benthic and pelagic morphs of Icelandic Arctic charr (*Salvelinus alpinus*). It may provide context in which the need to assessing the development of these morphs may fit.

We added this reference to the Introduction.

References.

Coyne JA & Orr AH. 2004. *Speciation*. Sunderland, Massachusetts: Sinauer Associates, Inc.

Footo AD. 2017. Sympatric Speciation in the Genomic Era. *Trends in Ecology and Evolution* **33**: 85–95.

Fridrich A, Hazan Y & Moran Y. 2019. Too many false targets for microRNAs: Challenges and pitfalls in prediction of miRNA targets and their Gene Ontology in model and non-model Organisms. *BioEssays* **41**: 1800169.

Horta-Lacueva QJB, Ólafsdóttir JH, Finn F, Fiskoviča E, Ponsioen L, Cámara M & Kapralova KH. 2022. From drones to bones: Assessing the importance of abiotic factors for salmonid spawning behaviour and embryonic development through a multidisciplinary approach. *Ecology of Freshwater Fish*: 1–11.

Irwin DE. 2020. Assortative mating in hybrid zones is remarkably ineffective in promoting speciation. *American Naturalist* **195**: E150–E167.

Nosil P. 2012. *Ecological Speciation*. Oxford: Oxford University Press.

Skúlason S, Snorrason SS, Noakes DLG, Ferguson MM & Malmquist HJ. 1989.

Segregation in spawning and early life history among polymorphic Arctic charr, *Salvelinus alpinus*, in Thingvallavatn, Iceland. *Journal of Fish Biology* **35**: 225–232.

Reviewers' comments:

Reviewer #1 (Remarks to the Author):

In this revised manuscript the authors have done overall a good job of answer the comments from both reviewers. For the record, I agree with the authors in their replies to reviewer 2.

Concerning the issue I raised about numbers of replicates, I am sorry to say that I do not find the reply and new table entirely clear. Pooling samples gives 1 sample, so how many replicates are obtained from the 9 embryos in this sentence?

"3 pooled families with 3 sampled embryos in each for each cross type-time point category"

If e.g. 9 embryos are sampled and then pooled 3 by 3, there are in the end only 3 replicates, even if 9 embryos were sampled.

This point needs to be very clear, maybe by adding information about pooling of RNA (if any) in Table 3.

Reviewer #2 (Remarks to the Author):

I have read the revised version of the manuscript 'Evolution of canalization: lessons from a classic case of resource polymorphism' and have some concerns with the data and analyses that have been performed in this version of the manuscript. I think the idea and theory within this manuscript is highly valuable and would make a great paper however, there are a few methodological issues that I think need to be addressed with the manuscript in it's current state. Please see the attached PDF document.

Evolution of canalization: lessons from a classic case of resource polymorphism

Response to Referees – Revision round 2

Referee 1.

In this revised manuscript the authors have done overall a good job of answer the comments from both reviewers. For the record, I agree with the authors in their replies to reviewer 2.

Concerning the issue I raised about numbers of replicates, I am sorry to say that I do not find the reply and new table entirely clear. Pooling samples gives 1 sample, so how many replicates are obtained from the 9 embryos in this sentence?

"3 pooled families with 3 sampled embryos in each for each cross type-time point category"
If e.g. 9 embryos are sampled and then pooled 3 by 3, there are in the end only 3 replicates, even if 9 embryos were sampled.

This point needs to be very clear, maybe by adding information about pooling of RNA (if any) in Table 3.

We are pleased to read that Referee 1 is satisfied with the work we have done in revising the manuscript.

We recognize that the sampling design could be better explained. We have improved this description by removing Table 3, and by replacing it with a diagram (new Fig 5). This visual depiction now clarifies that pooling was made at the family levels and was only statistical (i.e., we considered the embryos of the same cross type x time point altogether regardless of the family). Note that this was only done for the LCV analyses since the linear model used for DE enables to fit the families as a fixed effect.

Fig 5. Sampling design for LCV analyses. Each cross-type x developmental time point category contained 9 embryos from 3 pooled families. Embryo icons by DBCLS <https://togotv.dbcls.jp/en/pics.html>, CC-BY 4.0; adult charr X. Giroux-Bougard, phylopic.org.

Referee 2.

I have read the revised version of the manuscript 'Evolution of canalization: lessons from a classic case of resource polymorphism' and have some concerns with the data and analyses that have been performed in this version of the manuscript. I think the idea and theory within this manuscript is highly valuable and would make a great paper however, there are a few methodological issues that I think need to be addressed with the manuscript in its current state. Here are my general comments and concerns:

1. The writing in the manuscript has been greatly improved from the last version, especially in the introduction and discussion sections. I would either get rid of paragraph 3 (lines 79-83) as it doesn't really add much to the background of the study or add to it substantially as it feels like a half written paragraph currently. I would switch the order in which paragraphs 4 and 5 are presented to make the introduction flow a bit better. I really like the use of figure 1 to frame hypotheses and predictions in the expression profiles.

We acknowledge that Paragraph 3, as it was written in the previous version, might have appeared like a floating statement, but it does contain important information since it is the part of the Introduction establishing the rationale on the expected effects of hybridization on canalization. Therefore, we kept this paragraph but added contextual information as well as connective sentences, so that it is now better integrated into the Introduction.

2. I have a few concerns with the quality of the data presented in this manuscript which may or may not be easily corrected with more explanation. First, there needs to be a lot more information on how the mRNA and miRNA data was processed after sequencing. Using FASTQC is great but we need to know basic descriptive statistics about things like how much missing data there is or what cut-offs that were used to filter the data? We need to know statistics like this to get an understanding of the quality of the data. Second, how good was the alignment to the *Salvelinus* genome? We need to know how well the sequences aligned to the genome to assess the quality of the data. The authors need to show the readers that the data is high quality and the genes they're assessing are where they're supposed to be. Third, the Arctic charr genome is a duplicated (tetraploid) genome and does not behave the way that a diploid genome does. The Arctic charr genome is full of paralogous genes that may be different in their gene expression profiles and affect the reliability of gene expression data. How have the authors assessed and accounted for the different paralogues of genes within the genome? Different paralogues can be hard to distinguish and to differentiate from each other but, can have large implications to the quality of the underlying data and the reliability of the downstream analyses.

We added explanations on the quality of the data and our decisions made during the pre-processing steps:

- We added information on filtering. Specifically, we indicated that we discarded genes with less than 10 reads prior DE analyses (line 561).
- We added information on alignment quality for mRNAs with STAR (lines 514-515). We also loaded the STAR log files to the GitHub script repository, so the reader can access all the information on the quality of the alignment for each sample. This is now stated in the Data availability section (line 620).
- We added the information on miRNA sequence alignment from mirDeep2 (lines 521-525).
- We explained our decision to discard one sample (PLxSB 200ts) with failed sequencing (lines 527-528).

Referee 2 also stated that many paralogous genes resulting from the full genome duplication in Arctic charr may be different in their gene expression profiles and therefore influence our results. We don't have strong reasons to consider this genome duplication as a concern. First, as it is now clear from the new information in the Methods, ~84% of the mRNA reads had a unique alignment on the *Salvelinus* genome, so we are confident that most reads were assigned to paralogs of the right linkage group (note with the STAR options did not include multimapping). FeatureCount was set to count reads at the gene level, so the paralogs annotated as such in the *Salvelinus* genome are well distinguished during the analyses (l. 512-515). Regarding miRNA, genome duplication is a minor concern relative to the other homology-generating processes, since miRNAs occur as repeated copies

over the genome. In the worst case, any bias in gene expression because of homology would remain consistent among all cross types (because all individuals have the same level of ploidy).

3. I think that the analysis of local coefficient of variation presented in this study is a really neat and interesting analysis. In the revision (line 119), the authors state that the window size used is 500 genes. First, I would like to see the authors use a wide range of window sizes to test to see how robust their results are. The window size of 500 genes seems like it was picked very arbitrarily and doesn't have a biological or statistical reason behind it. Second, I would like to know how the authors accounted for gene density across the genome. Genes are not distributed evenly across the genome or even along a single chromosome and gene density is known to play a role in evolution. In the following diagram, in windows 1 and 2 of A) the genes (red squares) are close together and may be apart of the same genetic module. Their relative expression would be dependent on each other. Here it makes sense to do the analysis as it's currently done. In B) window 3 contains genes that are far apart from each other and not apart of the same genetic module. Their relative expression patterns would not be dependent on each other and it wouldn't really make sense to assess variability in expression based on genes within this window. Without accounting for gene density, the window may be in either a very large part of the genome (window 3 in B) or a very small part of the genome (windows 1 and 2 in A). A potential solution to get around this issue would be to pick a window size based on the average gene density for each chromosome individually. Each chromosome will vary in the number of genes present and this might help to get more accurate results rather than using a flat (and arbitrary) window size of 500 genes. Lastly, if the authors have not accounted for paralogues in the filtering or analyses then the expression of these paralogous genes may also affect the variability results within this analysis.

We are pleased to read that Referee 2 appreciates the interest of our analyses on local coefficient of variation. Referee 2 requests clarifications regarding the choice of parameters.

We applied the default window size provided by Simonovsky and colleagues 2020 (doi: 10.1093/bioinformatics/btz023). As shown in this methods paper, window sizes between 100 and 1000 provide highly similar results, and even extremely small (10) or large (10 000) window sizes provide robust LCV estimates. The setting choice for this parameter is purely statistically motivated.

We suspect that Referee 2 understood that the algorithm processes genes sorted by genomic coordinates, which is not the case (the genes are sorted by expression levels). Our analyses are therefore agnostic to the genomic architecture and to spatial variations in gene expression. Thus, we disagree that picking a window size based on gene density is appropriate. However, we acknowledge that the reader can easily be mistaken about the way genes are sorted, so we clarified this point in the methods (lines 538-539).

4. The results in figure 2 and 3 from the local coefficient of variation analysis appear to be clustered into 10 groups. How were these clusters made? It again seems like the authors picked an arbitrary number of clusters to group the data into without a biological reason. Was the clustering supervised or unsupervised and how do the clusters relate to each other? The figures appear to have a hierarchical structure to them but this is not explain at all in the methods. The methods section should be clear, descriptive, and allow your study to be completely repeatable not vague. Why not

show the results from this analysis in a manhattan plot fashion? It could be really interesting to see how the LCV results varies across the genome. Is it specific to certain chromosomes, does gene density come into play with this analysis, are there hotspots and cold spots of expression variation? Loads of questions could be asked. At least put it into the supplementary.

We acknowledge that the technical specificities of these heatmaps were very briefly explained. We detailed this information further in the methods (l. 544-549). The heatmaps were produced by unsupervised clustering. We extracted 10 clusters of genes by referring to the grouping of tips in the horizontal dendrogram. We verified the optimality of this choice by comparing the average silhouette widths of the same dataset partitioned into up to 50 clusters. The related figure was added to the supplementary material (Fig. S2). With this figure, one can see that partitioning the mRNA dataset further does not provide more information. Partitioning the miRNA dataset into more clusters could have provided slight informational gains, but an unrealistic partitioning for the downstream analyses (> 50 clusters) would be required. Therefore, taking 10 clusters is a good trade-off between informational content and simplicity.

Fig S2. Optimal number of clusters for the miRNA dataset. Average silhouette widths for (a) the mRNA dataset and (b) the miRNA dataset.

Referee 2 also suggests presenting the data as a Manhattan plot and providing more analyses on spatial LCV variations over the genome. While these are certainly very interesting aspects of gene expression variability, we wish to focus on our initial questions: whether gene expression variability diverges rapidly, and what are the effects of hybridization on such pattern of divergence. Presenting the LCV results as a heatmap enables to easily depict the distances between cross types in these estimates (and therefore to assess patterns of dominance, maternal effects or transgressive variance in hybrids). Establishing the proximal causes of the observed gene clusters (e.g., genomic distance, gene network topology, homologs resulting from the full genome duplication) would greatly further

our understanding of the evolution of canalization, but are the matter of several follow-up papers. We are pleased to read that Referee 2 finds much potential in our data, and we acknowledge that it is often frustrating to not see sequencing data not been exploited in a way to tackle very classical aspect in descriptive genomics, but doing so would blur the main message of our paper. Likewise, our supplementary file contains many tables and figures to help understanding the data and to facilitate reproducibility, and we do not think that it would be appropriate to turn this additional file into a companion paper.

5. The results from table 1 pretty clearly show evidence of maternal effects at the 200t developmental time point. Both of the statistics are significant and show the effects of maternal effects in gene expression. There was a really interesting result which wasn't mentioned in the manuscript. There was always greater differential expression of mRNA when the PL was the mom in the hybrid pairing. This may indicate that the PL morph has a greater maternal effect or maternal influence in the development that the SB morph. Which would be a little counter intuitive to the conclusions made in the discussion. I thought it was a neat result in the table that could be drawn upon.

Referee 2 noticed the importance maternal effects and of what we referred to as "limnetic biases in expression" in our results. These maternal effect and morph biases are described in two paragraphs in the Results (lines 242-260). We also referred to this pattern of the Discussion in lines 371-376 et 407-425. We recall that our analyses on *average* gene expression are there to contextualise our main findings on LCV, so we would like to remain synthetic in our treatment of these aspects.

Specific comments

Line 52: The authors state there are strong reasons for canalization to occur during the speciation process. I do not disagree with this statement, but could they provide some of the reasons? Just to provide some context for readers who may not be familiar with the process of trait canalization.

This is the opening paragraph of the discussion, the rationale on the reasons for canalization to evolve during speciation is detailed in the next three paragraphs.

Line 53-54: The authors are correct, a lot of work has been focused the canalization of phenotypic traits. This would be a great place to state why it may be beneficial to focus on gene expression or the variability of gene expression profiles. This might help the get at the novelty of this study more effectively.

Explanations on the relevance of gene expression variability for canalization studies are to be found below this opening paragraph, in paragraph 5.

Line 466: Why did you sample the embryos at those two very specific developmental time points? What's the biological reason behind that?

We agree that more details on the relevance of these time points can be provided. We added explanations on the developmental meanings of these time points in the Methods section, lines 480-484. To us, the information in Introduction (lines 143-144) provides enough information on the

developmental relevance of our experimental design without being too technical, so we wish to keep it as such.

Line 467: Please say that you have 9 embryos per cross type for each time period to make this as clear as possible.

We improved this statement accordingly (now line 480).

Line 472. Need a period between temperature and Sample.

We added a period.

Line 528: Why did you use different parameters in the glm models for the mRNA and miRNA datasets? Is there a biological or statistical reason, if so please state that.

This paragraph in the Methods refer to different number of iterations, thinning interval and burnin, which are not statistical parameters in the narrow sens but are only used to optimise convergence. MCMC models with different data are not expected to converge at the same moment. Iterations, thinning interval and burnin are set iteratively by running MCMCs several with increasing values until satisfying diagnostic plots and effective sample sizes and obtained. This is common practice for this type of models, and we are convinced that readers wanting to replicate these particular analyses are aware of it.

Figure 2 and 3: I'm confused on why there are only 10 clusters. Are these clusters the result of the 500 gene sliding window analysis? First off, what analysis was done to get the hierarchical or phylogenetic nature of the clusters? Is each horizontal row must be the expression variability of each individual gene within the cluster. Are the clusters related to each other and if so how? Why do different clusters have different numbers of genes within them? If you used a flat window size of 500 genes across the entire arctic charr genome then each cluster would be the same size! How are there only 10 clusters across all 39 chromosomes in the Arctic charr genome?? Why not plot this link a manhattan plot using a sliding window analysis?

Our explanations on how the clustering was made are in our response to the general comment 3.

Regarding Referee 2's question on how clusters are related to each other: Genes are clustered based on LCV covariance among "cross type x time point" groups. The relatedness of each gene (and thereby of each clusters including those) is depicted with the horizontal dendrogram on the left of each heatmap.

Regarding Referee 2's question on why different clusters have different number of genes: Referee 2 mentioned that "If we used a flat window size of 500 genes across the entire Arctic charr genome then each cluster would be the same size". This is wrong: the sliding window (which is only used to correct the variances estimates for the reads number of each gene) does not impose any partition to the dataset, and the algorithm returns a single dataframe of LCV estimates with no retained information on the sliding window (because it is unimportant and irrelevant for downstream analyses). We suspect that Referee 2 confuses the algorithm used here with others that rely on a sliding window for genome scan analyses. We did not imply anywhere in the manuscript that the

number of clusters could be in any way directly related to size of the sliding window. To avoid confusion, we made explicit that the clustering relies on LCV covariance among cross types 544-548.

We disagree with Referee 2's suggestion of presenting these results with a Manhattan plot, as written in our answer to the general comment 4.

Figure 3: The different crosses only differ in miRNA variability in clusters 6, 8, and 9.

We disagree with Referee 2 because nonoverlapping 95 % credible intervals between cross types are also observed in cluster 5,7 and 10. Also, partially overlapping credible intervals conferring a modest level of statistical confidence are found in cluster 4.

Figure 4: Panel A) and B) need to be flipped so that panel A) is on the left and panel B) is on the right. Also the second part of each panel should go into the supplementary as it's not really needed.

First, we would gently ask Referee 2 to remain considerate with peers during this correspondence.

Figure 4 appeared vertically in the initial submission. By flipping it back into landscape view, as presented here below, one can see that Panel (a) is well positioned to the left of Panel (b). We understand this graph may appear quite complex at first sight, but Upset graphs are common visual tools to depict shared variations among many groups (https://en.wikipedia.org/wiki/UpSet_Plot). Upset graphs are simple alternatives to Venn plots when more than three groups are compared. In our case, by reading the Upset plots in their entirety one can directly read how many genes are differentially expressed between cross type and how many of those genes are unique to each comparison, so this Figure gives essential information on both differential expression, maternal effects and morph-biases. Therefore, we wish to keep Figure 4 as it is.

Fig. 4. Intersections between the sets of differentially expressed mRNAs in each cross type comparison, at (a) $150\tau_s$ and (b) $200\tau_s$.

Lines 238-239: There has been very little developmental biology mentioned this far into the manuscript. What do these specific time points correlate with? This should be in the introduction.

We added more information on the developmental relevance of these time points as answered above.

Lines 245-251: The statements made by the author aren't quite right and they don't mention a really neat result that would push this paper a bit further.

Referee 2 alleged that this statement isn't quite right without clearly justifying why. We are aware that GO analyses are informative to a certain extent only, but we judge that including a minimum of

contextual information on the putative functions of the genes involved in our observed patterns of variation owes to remain in the manuscript.

Line 400: Brought not broug

We corrected this wording.

REVIEWERS' COMMENTS:

Reviewer #1 (Remarks to the Author):

The new figure 5 answers my concerns about replicates and pooling. I have no further comments and look forward to seeing this paper published.

Reviewer #2 (Remarks to the Author):

In this revised version of the manuscript the authors have done a good job with revising the manuscript. I want to thank the authors for putting up with all of the questions and suggestions that I had on the last review. I think the manuscript is a good addition to the literature on canalization. I have no further questions or suggestions for the authors.